Weight-dependent susceptibility of tilapia to tilapia lake virus infection

Roy Sri Rajiv Kumar
Yamkasem Jidapa
http://orcid.org/0000-0002-9722-5569 Tattiyapong Puntanat
http://orcid.org/0000-0002-5707-3476 Surachetpong Win fvetwsp@ku.ac.th
Department of Veterinary Microbiology and Immunology, Faculty of Veterinary Medicine, Kasetsart University, Kasetsart University , Bangkok , Thailand
Nowak Barbara
Electronic publication date: 2021 Jul 6
Publication date: 2021
Volume: 9
Electronic Location ID: e11738
Received 2021 Mar 31; Accepted 2021 Jun 17
Copyright: © 2021 Roy et al.
Copyright year: 2021
Copyright holder: Roy et al.
License: This is an open access article distributed under the terms of the Creative Commons Attribution License, which permits unrestricted use, distribution, reproduction and adaptation in any medium and for any purpose provided that it is properly attributed. For attribution, the original author(s), title, publication source (PeerJ) and either DOI or URL of the article must be cited.
License URL: https://creativecommons.org/licenses/by/4.0/

Keywords: Tilapia lake virus, TiLV, Tilapia, Weight, Pathology

Funding: Kasetsart University Research and Development Institute FF(KU) 25.64 The Graduate School, Kasetsart University This research is supported by the Kasetsart University Research and Development Institute under project number FF(KU) 25.64 and by the Graduate Program Scholarship from The Graduate School, Kasetsart University. There was no additional external funding received for this study. The funders had no role in study design, data collection and analysis, decision to publish, or preparation of the manuscript.

==============================
The emergence of tilapia lake virus (TiLV) has had a severely negative impact on global tilapia aquaculture. TiLV infection has been reported at different life stages of tilapia, with more emphasis on fry and fingerlings; however, the virulence and pathology of TiLV at different tilapia size remains unexplored. In this study, tilapias from a single spawning were raised to 5 g, 25 g, and 65 g, and subsequently challenged by the intraperitoneal injection and cohabitation of a virulent strain of TiLV. The cumulative mortality, viral load, and histopathology of the fish were determined until 22 days post-infection (dpi). The cumulative mortality of the 5 g, 25 g, and 65 g fish was 85% (±1.67), 55% (±2.89), and 51.67% (±7.49), respectively. At 14 dpi, the mean TiLV load in the liver of the 5 g fish was significantly higher than in the 25 g and 65 g fish. All the weight groups showed severe pathological changes in the liver, spleen, and intestine after TiLV infection, but no particular difference was otherwise noted during the study with the exception of higher pathological scores in the liver of the small fish at 14 dpi. Overall, this study indicated that small fish are more susceptible to TiLV infection than large fish. Although multiple factors, including environmental factors, farm management practices, strains of virus could contribute to different susceptibility of fish to viral infection, the present study provides the evidence to support that fish weight affects the mortality and clinical outcome during TiLV infection. More intensive measures such as strict biosecurity and disease surveillance during the susceptible weight should therefore be emphasized to reduce the impact of this virus.

Introduction

Tilapia is the second most cultured fish species worldwide, with an annual production of 6.4 million tons and a projected value of 9.8 billion USD (FAO, 2017). The popularity of tilapia aquaculture has expanded rapidly due to the tilapia’s affordability, status as a high-quality protein source, strong disease resistance, and easy adaptation to adverse environments. However, the recent detection of tilapia lake virus (TiLV) is having a significant impact on tilapia production (Eyngor et al., 2014; FAO, 2017; World Organisation for Animal Health (OIE), 2018; Surachetpong, Roy & Nicholson, 2020). The virus causes high mortality in tilapia up to 90% in cases of natural infection and is responsible for the immensely negative economic impact on tilapia production in several countries (Fathi et al., 2017; Surachetpong et al., 2017). TiLV has been identified in 16 countries across four different continents (Surachetpong, Roy & Nicholson, 2020). A recent genomic analysis characterized TiLV in the new genus Tilapinevirus and species Tilapia tilapinevirus under the family Amnoonviridae and order Articulavirales (Adams et al., 2017).

The life stage or weight of fish at the time of exposure to pathogens is an important factor influencing mortality (Bergmann et al., 2003; Jaramillo, Hick & Whittington, 2017; Sollid et al., 2003). In general, juvenile fish are more susceptible to viruses, bacteria, or parasites than adult fish. For instance, most strains of infectious hematopoietic necrosis virus (IHNV) cause high mortality in small (2–20 g) rainbow trout (Oncorhynchus mykiss) than large (50 g) fish (Bergmann et al., 2003). Moreover, subclinical infection of nervous necrosis virus (NNV) in barramundi (Lates calcarifer) occurs in fish at 5, 7, and 9 weeks, while mass mortality and more severe clinical signs develop in small fish at 3 to 4 weeks, suggesting the impact of the age of the host during exposure to pathogens (Jaramillo, Hick & Whittington, 2017). There have been no detailed studies on the weight or life stage-related susceptibility of tilapia to TiLV although it has suggested that all life stages of tilapia, including fertilized eggs, fry, juveniles, adult, and brood stock, are prone to TiLV infection (Dong et al., 2017; Yamkasem et al., 2019). It has been reported that during field outbreaks, juvenile fish and fingerlings at the weights 1–10 g are more susceptible to infection than adult fish (>100 g) (Eyngor et al., 2014; Jansen, Dong & Mohan, 2019; Tattiyapong, Dachavichitlead & Surachetpong, 2017). Likewise, high mortality (20–90%) associated with TiLV infection has been described in small fish (1–50 g) (Surachetpong et al., 2017), while low mortality (9.2%) has been observed in adult tilapia (Fathi et al., 2017). In laboratory challenge studies, mortality ranging from 45% to 100% has been recorded in juvenile Nile tilapia and red hybrid tilapia at 10–15 g (Behera et al., 2018; Eyngor et al., 2014; Liamnimitr et al., 2018; Tattiyapong, Dachavichitlead & Surachetpong, 2017). In the present study, the impact of weight on the susceptibility of tilapia to TiLV was investigated. The cumulative mortality, viral load, and pathology after intraperitoneal (IP) injection and cohabitation challenge by TiLV were examined. Our findings suggest that fish weights influence the outcome during TiLV infection.

Materials and Methods

Animals and experimental designs

In total, 400 red tilapia hybrid (Oreochromis spp.) with an initial body weight of 1.0 ± 0.1 g were acquired from a tilapia hatchery with a TiLV-free status from Petchaburi province, Thailand. The fish were kept in the animal research facility of the Faculty of Veterinary Medicine at Kasetsart University, Bangkok, Thailand, in 400 L tanks at 28 °C with daily water exchanges up to 50%. After 1 week of acclimatization, five fish were randomly selected for screening for important pathogens, including TiLV by reverse transcription-quantitative PCR (RT-qPCR), bacterial isolation by anterior kidney sampling, and parasites by skin scraping and gill excision. Fish were fed with commercial feed three times per day until reaching an average size of 5 g, 25 g, and 65 g. The average age of 5 g, 25 g, and 65 g fish were 8, 12, and 16 weeks, respectively. At the same density (5 g/L), sixty fish from each weight group were equally divided into two 150 L tanks (for 5 g fish), three 150 L tanks (for 25 g fish), and six 150 L tanks (for 65 g fish). These tanks were dedicated to recording mortality. A further 30 fish of 5 g, 25 g, and 65 g were placed in additional 150 L tanks at the same density for sample collection. For each weight group, an additional 15 fish were used as the control group. The animal use protocol was approved by the Kasetsart University Institutional Animal Care and Use Committee (protocol number ACKU63-VET-011).

Virus propagation and challenge study

The TiLV strain VETKU-TV08 isolated from red hybrid tilapia collected in 2019 was used in the challenge study. The virus was propagated in the E-11 cell line, which is a clone of SSN-1 from snakehead fish (Ophicephalus striatus). The E-11 cell line was purchased from the European Collection of Authenticated Cell Cultures (ECACC, Porton Down, Salisbury, UK). The E-11 cells were maintained in Leibovitz’s L-15 medium (Sigma-Aldrich, St. Louis, MO, USA) and supplemented with 5% (vol/vol) fetal bovine serum (Thermo Fisher Scientific, Waltham, MA, USA) and 2 mM L-glutamine. The cells were propagated at 25 °C without CO2. The infected E-11 cells were harvested through centrifugation at 3,000×g for 10 min, and the supernatant was then collected and stored at −80 °C for later use. Before the IP injection challenge, the fish were sedated with a 1 mL/L eugenol (Better Pharma, Bangkok, Thailand) solution for 3 min. The fish were IP-challenged with 50 µL of TiLV at 105 TCID50/mL or the L-15 medium for the control group. During the experiment, the decision criteria to terminate fish included the appearance of severe clinical signs with two or more appearance including stop feeding for 3 consecutive days, severe erratic swimming, skin erosion, skin hemorrhage, scale protrusion, extensive abdominal swelling, and exophthalmia. For viral quantification, the liver tissues were collected from the control fish (n = 3) at day 0 of all weight groups and TiLV IP-challenge fish (n = 7−8) at 7, 14, and 22 days post-infection (dpi). The fish were euthanized using an overdose of eugenol solution (3 mL/L) for 5 min prior to fish necropsy and sample collection. The samples were placed in separate 1.5 mL microcentrifuge tubes, and stored at −20 °C for further RNA isolation.

For the cohabitation study, 10 fish of 5 g and 25 g were IP-injected with 50 µL of the TiLV strain VETKU-TV08 at 105 TCID50/mL and then placed in a 150 L glass tank containing 30 fish (cohabitation fish) of either 5 g or 25 g fish, giving a ratio of inducer to cohabitant of 1:3 (Liamnimitr et al., 2018). All IP-injected fish (inducer) were trimmed on the pelvic fin to differentiate them from cohabitating fish. Clinical signs of infection and cumulative mortality were observed and recorded until 28 days post challenge. At 7 and 14 dpi, three cohabitation fish from the 5 g and 25 g groups were randomly euthanized to collect liver samples and for processing for RNA isolation.

RNA extraction and cDNA synthesis

The total RNA was extracted from the liver using GENEzol™ reagent (Geneaid Biotech Ltd., New Taipei City, Taiwan) according to the manufacturer’s instructions. Briefly, 30 mg of liver samples were mixed and homogenized in 1 mL of GENEzol™ reagent using a hand-held pestle homogenizer. Thereafter, 200 μL of chloroform was mixed and incubated at room temperature for 3 min. The samples were then centrifuged at 4°C and 12,000×g for 15 min. The supernatant was transferred to a new microcentrifuge tube, mixed with 500 μL isopropanol, and incubated at −20 °C for 2 h. The samples were centrifuged at 4 °C and 12,000×g for 15 min to precipitate the RNA. After discarding the supernatant, the RNA pellet was washed with 75% ethanol and centrifuged at 4 °C and 10,000×g for 15 min. The RNA pellet was resuspended in 50 μL prewarmed RNase-free water (60 °C). The RNA quality and quantity were examined using a NanoDrop™ 2000 spectrophotometer (Thermo Fisher Scientific, Waltham, MA, USA).

For cDNA synthesis, a 20 µL mix reaction containing 4 μL of 5X RT buffer mix, 1 μL of primer mix, 1 μL of RT enzyme mix, 4 μL of nuclease-free water, and 10 μL total RNA template (1 μg) was prepared using a reverse transcription kit (Toyobo, Osaka, Japan). The reaction was incubated in a T100 thermal cycler (Bio-Rad, Hercules, CA, USA) at 42 °C for 60 min, followed by 98 °C for 5 min.

Reverse transcription-quantitative polymerase chain reaction

The TiLV genomic RNA was measured using an SYBR-based reverse transcription-quantitative polymerase chain reaction (RT-qPCR) assay (Tattiyapong, Sirikanchana & Surachetpong, 2018). Briefly, the reaction was performed in a 10 μL reaction containing 4 μL of 400 ng cDNA, 5 μL of 2X iTaq™ universal SYBR supermix (Bio-Rad, Hercules, CA, USA), and 0.3 μL of forward and reverse primers. The final volume was adjusted to 10 μL using molecular water. The reactions were performed in a PCR thermal cycler, CFX96 Touch™ (Bio-Rad, Hercules, CA, USA). At the end of the qPCR reaction, the samples were processed for melting curve analysis at 65–95 °C with increments of 0.5 °C per 5 s. The TiLV viral concentration was extrapolated by comparing the Ct value of the tested samples to the standard curve generated from a 10-fold serial dilution of a plasmid-containing segment 3 of TiLV.

Histopathology

For the histopathological analysis, two control and three TiLV IP-challenge fish were collected from each weight group at 7, 14, and 22 dpi. Tissues, including the liver, spleen, and intestine, were removed and placed in 10% (vol/vol) neutral buffered formalin. At 24 h, the tissues were transferred to 70% ethanol. The samples were then processed using a standard histopathology protocol in which they were embedded in the paraffin block, sectioned at 5 µM thick, and stained with hematoxylin and eosin (H&E). Thereafter, the tissue slides were scanned using VS120® Virtual Microscopy Slide Scanning (Olympus, Tokyo, Japan) and examined and graded under an Olympus OlyVIA Ver.3.1 program (Olympus, Tokyo, Japan).

Statistical analysis

The difference in the cumulative mortality, and mean TiLV concentration in the IP-challenge and cohabitation experiments, pathological scores from each weight groups and different time points were determined using GraphPad Prism software version 5.0 (GraphPad, San Diego, CA, USA). Significant differences were assessed using one-way ANOVA with Tukey’s multiple comparisons test or nonparametric Mann–Whitney test (Gibson-Corley, Olivier & Meyerholz, 2013). A p-value less than 0.05 was considered significant.

Results

Susceptibility of small fish to TiLV infection

After TiLV infection, an earlier onset of clinical signs, including lethargy, anorexia, schooling cessation, and lying on the floor of the tank, was observed in the small fish (5 g) at 2 dpi, while these clinical signs started at 4 dpi in the medium (25 g) and large fish (65 g). More severe clinical signs and gross lesions, including exophthalmos, skin hemorrhage, scale protrusion, anemia, fin erosion, and ascites, were exhibited in all weight groups after 4 dpi (Fig. S1). At the end of the experiment, the mean cumulative mortality and standard error of the mean (SE) of the small, medium, and large red hybrid tilapia was 85% (±1.67), 55% (±2.89), and 51.67% (±7.49), respectively, with significantly higher mortality in the 5 g fish than the other groups (p < 0.05; Fig. 1). The small fish had first mortality at 3 dpi, while this occurred in the medium and large fish at 7 dpi. Notably, the mortality of the small fish continued until 16 dpi, while the mortality of the medium and large fish stopped at 13 and 14 dpi, respectively. No mortality or signs of TiLV infection were recorded in the control fish in any weight group at any time during the study.

Figure 1 Cumulative mortality of the 5 g, 25 g, and 65 g red hybrid tilapia after the TiLV challenge.

At the density of 5 g/L, sixty fish from each weight group were equally divided into two 150 L tanks (for 5 g fish), three 150 L tanks (for 25 g fish), and six 150 L tanks (for 65 g fish). Then, fish were intraperitoneally (IP) injected with 50 µL of TiLV at 105 TCID50/mL. Mortality data were recorded from individual tank and presented as a mean and SE from each weight group. The challenge studies was performed at 1, 2, and 3 months when fish were allowed to grow to the expected size in our animal facility until they reached 5 g, 25 g, and 65 g respectively. The clinical signs and daily mortality were observed and recorded for 21 days. Mortality data from different size of fish was compared using one-way ANOVA with Tukey’s multiple comparisons test. An asterisk (*) indicates a statistical difference (p < 0.05).

Amount of TiLV in small fish

At 7 dpi, the mean TiLV genomic RNA concentration in the liver of the 5 g, 25 g, and 65 g fish was 7.4 log10, 7.1 log10, and 6.7 log10 TiLV copy/µg total RNA, respectively (Fig. 2). In all weight groups, the mean viral load at 14 dpi was significantly lower at 5.7 log10, 2.7 log10, and 3.1 log10 TiLV copy/µg total RNA in the 5 g, 25 g, and 65 g fish, respectively. No difference in the mean viral load was found at 22 dpi (3.4 log10, 3.2 log10, and 3.0 log10 TiLV copy/µg total RNA in the 5 g, 25 g, and 65 g fish, respectively). From the three time points, the 5 g fish showed more TiLV genomic RNA in the liver than the 25 g fish (p < 0.001) and 65 g fish (p < 0.01) at 14 dpi (Fig. 2). No TiLV genomic RNA was detected in any of the weight groups at 0 dpi before the TiLV challenge.

Figure 2 Comparison of TiLV RNA concentrations in TiLV-IP challenge fish.

The amount of TiLV RNA was analyzed from the liver of 5 g, 25 g, and 65 g fish (n = 8) at 7 dpi and fish (n = 7) at 14 and 22 dpi. The liver of fish (n = 3) were collected from the 5 g, 25 g, and 65 g groups at 0 dpi prior to the TiLV challenge to demonstrate the TiLV status of the fish. The viral concentration from different size of fish and different time points were compared using one-way ANOVA with Tukey’s multiple comparisons test. The asterisks indicate a statistical difference (**p < 0.01, ***p < 0.001).

Histopathological scores of the organs in the different weight groups

Histopathological changes in the liver, spleen, and intestine of all the weight groups were scored according to the scoring system described in Table 1 and Fig. 3. The scores were given after examining three fish per weight group per time point. The three categories of scores were normal (0), mild (1), moderate (2), and severe (3) (Fig. S2). At 7 dpi, all the weight groups obtained severe pathological scores in the liver, spleen, and intestine. At 14 dpi, lower pathological scores were given for the liver, spleen, and intestine of the 25 g and 65 g fish than the 5 g fish. In particular, most of the 5 g fish had moderate or severe pathological scores in these organs compared to the 25 g and 65 g fish (Table 2). Low histopathological scores were obtained in all weight groups at 22 dpi (data not shown) and no pathological changes in the control unchallenged fish.

Table 1 Histopathology scoring system.

Score	Severity	Tissue	
Liver	Spleen	Intestine	
0	Absent	normal	normal	normal	
1	Mild	10–20% liver cells degeneration and necrosis	10–20% red blood cell depletion	10–20% infiltration of lymphocyte in lamina propria	
		0–3 syncytial cells		
		0–3 intracytoplasmic inclusion bodies			
2	Moderate	20–40% liver cells degeneration and necrosis	20–40% red blood cell depletion	10–20% infiltration of lymphocyte in lamina propria	
		20–40% depletion of glycogen in hepatocytes	10% increase melanomacrophage centers	
		4–10 syncytial cells	1–5 intracytoplasmic inclusion bodies	10–20% sloughing of intestinal epithelial cells	
		4–7 intracytoplasmic inclusion bodies		20–40% goblet cell hyperplasia	
3	Severe	>40% liver cells degeneration and necrosis	>40% red blood cell depletion	10–20% infiltration of lymphocyte in lamina propria	
		>40% depletion of glycogen in hepatocytes	>10% increase melanomacrophage centers	
		>10 syncytial cells	>5 intracytoplasmic inclusion bodies	>40% sloughing of intestinal epithelial cells	
		>7 intracytoplasmic inclusion bodies		>40% goblet cell hyperplasia	

Figure 3 Histopathology score of liver, spleen, and intestine of normal and TiLV-IP challenge fish.

Representative histopathology of (A) Liver. (E) Spleen. (I) Intestine of normal fish. (B) TiLV-IP challenge fish showed degeneration and necrosis of hepatocytes, and depletion of glycogen in hepatocytes. (C) Syncytial hepatic cells (arrow head). (D) Eosinophilic intracytoplasmic inclusion bodies (arrow). (F) The spleen of TiLV-IP challenge fish showed red blood cell depletion. (G) Increased melanomacrophage centers. (H) Eosinophilic intracytoplasmic inclusion bodies (red arrow). (J) The intestine of TiLV-IP challenge fish showed infiltration of lymphocyte in lamina propria. (K) Sloughing of intestinal epithelial cells (L) Goblet cell hyperplasia. Histopathological scores in organs of each weight group (n = 3) were evaluated (M) at 7 dpi. (N) at 14 dpi.

Table 2 Pathological scores of liver, spleen, and intestine in TiLV-challenge tilapia.

Fish weight	Organs	Pathological scores*	
7 dpi	14 dpi	
Scores	Number of affected fish/total fish	Scores	Number of affected fish/total fish	
5 g	Liver	3	2/3	3	2/3	
		2	1/3	1	1/3	
	Spleen	3	3/3	3	2/3	
				2	1/3	
	Intestine	3	2/3	3	2/3	
		2	1/3	2	1/3	
25 g	Liver	3	2/3	2	1/3	
		1	1/3	1	2/3	
	Spleen	3	1/3	2	1/3	
		2	2/3	1	2/3	
	Intestine	3	1/3	3	1/3	
		2	2/3	1	2/3	
65 g	Liver	3	2/3	2	1/3	
		2	1/3	1	2/3	
	Spleen	3	2/3	3	1/3	
		1	1/3	2	1/3	
				1	1/3	
	Intestine	2	1/3	3	1/3	
		1	2/3	2	2/3	
Note:

* The pathological scores were categorized as mild (1), moderate (2), and severe (3) changes according to the scoring system mentioned in Table 1. The scores were categorized after examing three fish per weight group per time point and statistically analyzed by nonparametric Mann–Whitney test.

Weight susceptibility to TiLV in the cohabitation challenge

The effect of weight on TiLV susceptibility was further tested in the cohabitation challenge study. As shown in Fig. 4, the cumulative mortality of the inducer 5 g and 25 g fish were 85% and 60%, respectively, while the mortality in the cohabitation 5 g and 25 g fish was 38% and 23%, respectively. Both the inducer 5 g and 25 g fish started showing clinical signs of TiLV infection on days 3–4, with the first mortality observed on 7 dpi. Notably, the cohabitation 5 g and 25 g fish had delayed clinical signs and mortality onset, which started on 9–11 dpi. Interestingly, while the mortality in the 5 g fish ceased at 20–24 dpi, the mortality of the 25 g fish stopped earlier, at 15–18 dpi (Fig. 4). Further analysis of the TiLV concentrations in the liver of the cohabitation 5 g fish at 7 and 14 days showed the viral load between 3.50 log10 and 5.64 log10 TiLV copy/µg total RNA. The mean viral load in cohabitation 25 g fish ranged between 2.86 log10 and 5.71 log10 TiLV copy/µg total RNA.

Figure 4 Cumulative mortality of the 5 g and 25 g red hybrid tilapia after the TiLV cohabitation challenge.

Fish (n = 30) from each group were cohabitated with an inducer (n = 10). Clinical signs and daily mortality were observed and recorded for 28 days. Mortality data from 5 g and 25 g fish was compared using one-way ANOVA with Tukey’s multiple comparisons test. An asterisk (*) indicates a statistical difference (p < 0.05). An asterisk (*) indicates a statistical difference (p < 0.05).

Discussion

Since 2014, TiLV has had severely negative impacts on global tilapia aquaculture (Surachetpong, Roy & Nicholson, 2020). To overcome the negative impacts of TiLV disease, it is necessary to identify the associated risk factors and to implement appropriate interventions. In this study, our results revealed that small tilapia are more susceptible to TiLV infection than large tilapia. Specifically, higher mortality and worse clinical signs were observed in the 5 g fish than the 25 g and 65 g fish. High mortality (above 50%) after TiLV infection has been observed consistently in experimentally challenged tilapia (Behera et al., 2018; Tattiyapong, Dachavichitlead & Surachetpong, 2017). In conditions of natural infection, TiLV can cause mortality ranging from 5% to 90% depending on multiple factors (e.g., co-infections with other pathogens and farm biosecurity practices) (Eyngor et al., 2014; Fathi et al., 2017; Nicholson et al., 2017; Nicholson et al., 2020; Surachetpong et al., 2017). A recent field outbreak investigation revealed high mortality (80%) in 10 g tilapia, with lower mortality (50%) recorded in 120 g tilapia (Rao et al., 2021). Similar to TiLV, the life stage of the salmonid species plays an important role in their susceptibility to IHNV infection. For instance, small fish up to two months of age are more susceptible to IHNV than adult salmon (Lapatra, 1998). Likewise, Bergmann et al. (2003) reported lower mortality in 40–50 g rainbow trout (Oncorhynchus mykiss) than 2.5–3 g and 15–20 g fish after exposure to different isolates of IHNV. Moreover, the impact of fish age for viral susceptibility has been highlighted in common carp, salmonid species, cyprinid species and percid species against spring viremia of carp virus (Embregts et al., 2017; Emmenegger et al., 2016). Under field conditions, TiLV disease frequently occurs when tilapias are at 1–80 g, the period when the farmers often stock fish in the farms. As such, we focused our study on the susceptibility of TiLV infection in 5–65 g fish. Indeed, multiple factors, including environmental factors, farm management practices, age and genetic of fish, and strains of virus could contribute to different susceptibility of fish to viral infection. Our study provides the evidence as to support field observation that younger fish being more susceptible to TiLV infection.

Susceptibility due to the weight of tilapia during TiLV infection was further confirmed in this study using RT-qPCR and histopathological analysis. The 5 g, 25 g, and 65 g fish had high viral loads at 7 dpi, but this declined dramatically to low but detectable levels at 22 dpi. Significantly, higher TiLV concentrations were detected in the livers of the 5 g fish than the 25 g and 65 g fish at 14 dpi. In the small tilapia, severe histopathological changes were found in multiple organs compared to fewer lesions in the large tilapia. The severe pathological changes and extensive viral replication could have overcome the hosts’ immune systems, thus contributing to the high mortality of the 5 g fish. A correlation between high viral load and mortality was demonstrated in Nile tilapia after a TiLV challenge via an intragastric route (Pierezan et al., 2020). Additionally, a correlation between the histopathological changes and the level of viral load has been reported in Atlantic salmon (Salmo salar L.) after exposure to the piscine myocarditis virus (Timmerhaus et al., 2011). Overall, these evidences suggest that a high viral load and severe pathological alterations contribute to high mortality in small fish during virus infection.

To reflect natural infection and further validate the impact of fish size during TiLV infection, we produced TiLV infection in 5 g and 25 g tilapia through a cohabitation challenge model. Both the inducer and cohabitating 5 g tilapia showed higher mortality rates than the inducer and cohabitating 25 g tilapia. Conceivably, the resistance in the 25 g fish could be partly explained by its different immune functions, which could play an important role in the control of virus replication in adult fish. Various factors and mechanisms such as differences in innate immunity or host cell permissiveness are likely involved in weight-dependent susceptibility to TiLV infection. Prior to this study, a different expression of cytokines and immune-related genes was reported in tilapia during TiLV infection (Mugimba et al., 2020; Pierezan et al., 2020; Sood et al., 2021). For example, Mugimba et al. (2020) showed that the TiLV viral load and expression of immune-related genes were inversely correlated in the brain and spleen of infected fish. Alternatively, a mechanism to facilitating viral persistence in younger fish e.g., production of immunosuppressive cytokines, could play a role in suppressing the host response to TiLV. Nevertheless, further study is required to dissect the mechanisms of host immunity in different sizes of fish against TiLV. In addition to the different immune regulation, the challenge route, strain of the virus, and condition of the fish could affect the outcomes of challenge studies (Eyngor et al., 2014; Liamnimitr et al., 2018; Mugimba et al., 2019; Pierezan et al., 2020; Tattiyapong, Dachavichitlead & Surachetpong, 2017). Understanding the stage at which fish are susceptible to pathogens could lead to the appropriate implementation of control measures during critical periods in fish aquaculture. Such control measures, including the probiotic Bacillus spp. or immunostimulants, could be applied to promote the immune system of the host prior to TiLV exposure. The positive effects of probiotics was highlighted in a recent study, which showed that probiotic supplementation with Bacillus spp. in tilapia feed improved fish survival while reducing the TiLV load in the organs of fish (Waiyamitra et al., 2020).

Conclusions

Our study demonstrated that fish weight strongly influences the outcome of TiLV infection. High mortality, an abundant viral load, and severe pathological changes were found in the small fish rather than the large fish. The application of control measures such as supplementation with probiotics or immunostimulants during the life stage or weight when tilapia are most at risk of infection could therefore help farmers cope with the negative impacts of TiLV.

Supplemental Information

Supplemental Information 1 Author checklist.

Click here for additional data file.

Supplemental Information 2 Dataset.

Click here for additional data file.

Supplemental Information 3 Raw data for Tables 1 and 2.

Click here for additional data file.

Supplemental Information 4 Gross pathology of TiLV-IP challenge fish.

Representative figures of (A–C) 5 g red hybrid tilapia. (D–F) 25 g red hybrid tilapia. (G–I) 65 g red hybrid tilapia.

Click here for additional data file.

Supplemental Information 5 Representative histopathological lesions of liver, spleen, and intestine of TiLV-IP challenge fish.

(A, E, I) Normal liver, spleen and intestine of control fish. (B–D) Liver of TiLV-IP challenge fish with mild, moderate, and severe lesions. (F–H) Spleen of TiLV-IP challenge fish with mild, moderate, and severe lesions. (J–L) Intestine of TiLV-IP challenge fish with mild, moderate, and severe lesions. Lesion scores were graded according to criteria described in Table 1.

Click here for additional data file.

Additional Information and Declarations

Competing Interests

Author Contributions

Animal Ethics

Data Availability

The authors declare that they have no competing interests.

Sri Rajiv Kumar Roy conceived and designed the experiments, performed the experiments, analyzed the data, prepared figures and/or tables, authored or reviewed drafts of the paper, and approved the final draft.

Jidapa Yamkasem performed the experiments, analyzed the data, prepared figures and/or tables, authored or reviewed drafts of the paper, and approved the final draft.

Puntanat Tattiyapong performed the experiments, analyzed the data, prepared figures and/or tables, authored or reviewed drafts of the paper, and approved the final draft.

Win Surachetpong conceived and designed the experiments, performed the experiments, analyzed the data, prepared figures and/or tables, authored or reviewed drafts of the paper, and approved the final draft.

The following information was supplied relating to ethical approvals (i.e., approving body and any reference numbers):

The animal use protocol was approved by the Kasetsart University Institutional Animal Care and Use Committee (ACKU63-VET-011).

The following information was supplied regarding data availability:

Raw data are available as a Supplemental File.

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
