# Peer review of "Weight-dependent susceptibility of tilapia to tilapia lake virus infection"

_PeerJ, doi:10.7717/peerj.11738_

## Round 0.1 · original submission · Major Revisions

· Academic Editor

Major Revisions

While this is an interesting paper it does have a lot of shortcomings. I agree with the reviewer that most of the histological descriptions appear to be incorrect and the changes described are impossible to see. I suggest that higher magnification is used and the features mentioned are clearly indicated on the micrographs. Insets at higher magnification could be used. Unless the histology is well illustrated, the histopathological scoring in Table 1 is questionable.

Please clarify if the cumulative mortality in Fig. 1 was from one experiment with all fish sizes tested at the same time. If not it should be clearly indicated in the figure caption. Why were the tanks pooled - this information could be included in the caption of the figure. Please clarify in the figure caption what statistical analysis was done. Did you use survival analysis?

Please include information on how old were the fish at 5, 25, and 65 g should be indicated.

More data of other fish that are age-dependent susceptible to viral infection should be discussed.

Were the fish in cohabitation tagged to be able to recognise which were donors and which were naive?

All methods must be included in the Materials and methods section.
Please carefully consider and address all comments from the reviewers.

Reviewer 1 ·

Basic reporting

-

Experimental design

-

Validity of the findings

-

Additional comments

PeerJ Reviewing Manuscript 59569v1: Weight-dependent susceptibility of tilapia to tilapia lake virus infection
This work demonstrated weight dependent susceptibility of tilapia to TiLV. Three different fish weights were used in injection test while 2 different weights were for cohabitation. The results suggested the smallest size (5 g) was the most susceptible to TiLV.
Comments:
1. The main concern of the manuscript is the histopathological readings. Most of the descriptions are incorrect. In Fig 3, there was no sign of syncytial formation, cytoplasmic inclusion bodies, occasional congestion and hemorrhaging in the liver. No melanomacrophage centers (MMC) and pyknosis and karyorrhexis in the spleen. No hemorrhaging and congestion in the brain. No disruption of the gastric glands, hemorrhage in the lamina propria, and vacuolation in the intestines. Thus, the histopathological scoring in Table 1 is questionable making data interpretation regarding weight dependent susceptibility weaken/invalid.
2. Cumulative mortality in Fig. 1 gave an impression that the experiment was performed at the same time. It might be better to present data as independent set up. Additionally, mortality from replicate tanks should be presented (instead of a combine total fish of 60) and statistical analysis can then be done.
3. Information on fish age at 5, 25, and 65 g should be indicated.
4. More data of other fish that are age-dependent susceptible to viral infection should be discussed. Any case of age-independent? Spring viremia of carp virus?
5. In cohabitation, how to tell which fish were inducers or naïve?
6. Lines 197-199, it was confusing what was the data for 5 and 25 g fish at 7 and 14 days.
7. Sample at day 0 collected for qPCR should be in the method section. Were fish from control (no virus challenge) subjected to qPCR?
8. A 28-day experiment for cohabitation experiment should be indicated in the method section.
9. Lines 101-103 “For viral quantification, the samples were collected at 7, 14, and 22 days post-infection (dpi). For sample collection, the fish were euthanized using an overdose of an eugenol solution (3 mL/L) for 5 min.” should be read as “For viral quantification, the samples were collected at 7, 14, and 22 days post-infection (dpi). The fish were euthanized using an overdose of an eugenol solution (3 mL/L) for 5 min prior to fish necropsy and sample collection”
10. TiLV quantification in Fig. 2 was assayed from livers of three fish but why more than three dots shown in the graph.

·

Basic reporting

Good writing article, with interesting fundamental knowledge on the effects of fish weight/size to TiLV infection.

Experimental design

Good experimental design. However, some comments for improvement as stated below and in attached file.

Validity of the findings

Good data validity due to good experimental design. However, some comments for improvement as stated below and in attached file.

Additional comments

Fulfill the criteria of basic reporting, experimental design and validity of the findings. However, some comments for improvement as stated below and in attached file.

---

## Round 0.2 · Minor Revisions

· Academic Editor

Minor Revisions

Thank you for the revised manuscript, it is significantly improved, please address minor changes as requested by one of the reviewers.

Reviewer 1 ·

Basic reporting

-

Experimental design

-

Validity of the findings

-

Additional comments

The revised manuscript has been satisfactorily improved. Just only one point; In the abstract, ± sign should be for SE values in the parentheses.

·

Basic reporting

Please refer to the attached file

Experimental design

Please refer to the attached file

Validity of the findings

Please refer to the attached file

Additional comments

Overall, significant improvements have been made by the authors. However, minor additional info as follows should be made too:

In the abstract and discussion sections:
This study only did on 5, 25 and 65 g of fish. What happens if size bigger than that - until harvest size? Bigger size causes more economic losses, compared to smaller size - even smaller size more susceptible in this study. This might be the limitation of the study. Please indicate here and in the discussion.

In the discussion section:
Please scientifically discuss why smaller size is more susceptible compared to bigger size? Any scientific explanation? This should be stressed in the discussion, rather than just comparing with previous studies.

Please refer to the attached file.

---

## Round 0.3 · accepted · Accept

· Academic Editor

Accept

Thank you for making the requested changes.